# AI and Data Analytics in the Dairy Farms: A Scoping Review

**DOI:** 10.3390/ani15091291

**Published:** 2025-04-30

**Authors:** Osvaldo Palma, Lluis M. Plà-Aragonés, Alejandro Mac Cawley, Víctor M. Albornoz

**Affiliations:** 1Department of Mathematics, Universidad de Lleida, 73 Jaume II, 25001 Lleida, Spain; lluismiquel.pla@udl.cat; 2Department of Economics and Administration, Universidad Nacional Andrés Bello, Santiago 8370133, Chile; 3Agrotecnio CERCA Center, 191, Rovira Roure, 25198 Lleida, Spain; 4Department of Industrial and Systems Engineering, Pontificia Universidad Católica de Chile, Santiago 7820436, Chile; amac@ing.puc.cl; 5Department of Industrial Engineering, Campus Santiago Vitacura, Universidad Técnica Federico Santa María, Santiago 7650568, Chile; victor.albornoz@usm.cl

**Keywords:** milk production, machine learning, data analytics, neural networks, simulation

## Abstract

In our research, we explored how technology can help address the rising demand for bovine milk driven by global population growth. Through a review of 151 articles, we investigated the role of data analytics in dairy farms. Our findings underscore the importance of predictive analytics in accurately forecasting milk production and detecting diseases like mastitis and lameness in cows. While historical data remain crucial, the inclusion of real-time data is increasingly valuable. We highlight the potential for future research to integrate simulation tools with machine learning methods, offering promising avenues for improving dairy production practices.

## 1. Introduction

Data Analytics (DA) is a scientific discipline involving the analysis of large and complex data sets to learn, extract, and uncover hidden patterns, unknown correlations, market trends, and other useful information to support decision-making [1]. As part of traditional operational research (OR) methods, DA uses advanced analytical techniques such as machine learning, data mining, natural language processing, and time series to extract insights from data. In recent years, the progressive digitalization of dairy farms, along with the rest of society, has led to massive data availability, representing an opportunity to process data into information by applying DA methods and to make better and more informed decisions. DA methods can help to significantly increase milk revenue or reduce production costs associated with precision livestock feeding, reducing wastage and ecological footprint, detecting reproductive or health problems at early stages before they are diagnosed as disease, anticipating the treatment and generating significant animal welfare and benefits by reducing the use of medicines [2].

To date, farmers have benefited from genetic improvements and more efficient milk production in commercial herds, besides the gradual adoption of farm management systems, sensors, and other electronic devices to ease farm operation, data gathering, and analysis. This technological progress relies on the extensive use of data analysis and statistical methods, often dominated by descriptive methods or static dashboards, to keep track of the main key performance indexes (KPIs). The rise of big data (BD), Artificial Intelligence (AI), and machine learning (ML) applications are gaining relevance in decision support systems (DSS) for their capability of dealing with large amounts of data, covering many dairy farm aspects such as feeding, breeding, health, animal behavior, milking, and resource management [3]. Among these aspects, milk production predominates since it is the core dairy product. The importance of milk production is also expected to grow as demand for milk grows steadily along with the world population. The high level of milk production is associated with technological management, thanks to the implementation of BD and ML technologies to address traditional issues in dairy farms [4]. Because of that, developed countries generate a significant amount of dairy production, even though they have a smaller dairy cow population compared to underdeveloped or developing countries.

Scientific reviews play a crucial role in knowledge transfer, helping preserve previously generated data, identifying knowledge gaps, and preventing the duplication of efforts [5]. A scoping review aims to summarize scientific knowledge in a field of study using a broad foundation, with the goal of understanding the area of study, posing broad research questions, conducting a comprehensive search, and performing descriptive analysis. Previous reviews related to DA tools in the dairy farms had been focused on various topics, such as (i) lameness issues as revised by Qiao et al. [6], who considered more than 100 articles published up to 2021 and discussed smart techniques for lameness detection and behavior recognition (e.g., aggression as bad behavior), highlighting ML techniques like deep learning, support vector machine, K-nearest neighbours, and decision trees; (ii) the use of specific information technology and data management tools, revised by Akbar et al. [4], who covered articles published up to 2019 and concentrated on how the Internet of Things (IoT) is transforming smart dairy farming in the context of growing global demand for dairy products as noted by [7], but which does not delve into DA techniques; (iii) specific analytical methods, revised by Dongre and Gandhi [8], who covered articles up to 2014 considering the application of artificial neural network models to predict and optimize aspects like milk production, heat detection, and prediction of reproduction values; (iv) ML applications like in the review of Slob et al. [9], who conducted a review between 2010 and 2020 and focused only on ML applications to improve dairy farm management under a veterinary perspective, while broader applications of ML were revised by Shine and Murphy [3], who conducted a review of articles from 1999 to 2021 and embraced dairy production problems, identifying geographical origins, characteristics, evaluation metrics and methods; and (v) economic aspects in decision support systems for meat and milk production published between 2016 and 2022 and revised recently by Bang et al. [10].

The contribution of the present work is a scoping review providing a current and comprehensive view of DA in the dairy farming field, envisioning new applications of DA in the future. Although previous reviews have been conducted on various aspects of dairy farms, none have considered data analysis as a discipline that encompasses statistical methods, operations research (OR) and simulation, the treatment of uncertainty, the processing of historical and real-time data, the software used to implement the models, and the scope of decision support (strategic, tactical or operational). This work contributes to the field by providing a current and comprehensive overview of how AD tools are applied on dairy farms, thus filling a significant gap in the literature and offering new perspectives for future research and practical applications. We believe that the above aspects are important because (i) we will be able to know the contribution of DA so far to the development of this area in order to analyze data, learn from it, and be able to make predictions, considering, for example, the quality of current models; (ii) we will be able to see, which ML models have been the most widely used, understanding why researchers have made these choices, and identifying underexplored methods; (iii) we will be able to compare DA techniques and statistics, which methodologies have obtained better results, and show possible synergies or complementarities among them; (iv) the treatment of uncertainty (from deterministic to stochastic) considered in the DA models applied to the dairy farms will allow us to identify the way in which the inherent uncertainty of a real herd is considered in models; (v) knowing the software and data visualization tools will allow us to decipher the computer tools preferred by researchers to carry out their studies; (vi) knowing the underlying decision-making scope (strategic, tactical or operational) will help us understand the time horizon and who are the decision-makers to whom these tools are addressed; (vii) the type of analytics used (descriptive, predictive or prescriptive) will allow us to know the coverage of the decision-making process emphasizing descriptive statistics of the herd data, the estimation of performances and predictions or prevailing the aim that the models themselves deliver recommendations for decision-making; (viii) the level of use of historical or real-time data, in order to identify a possible gap in the use of real-time data that, for example, can be useful for timely identification of animal diseases and, thus, help optimize herd performance.

Thus, we are interested in answering the research question: How has DA been applied to dairy farms? To answer this research question, relevant studies on the application of DA in dairy farms will be selected through the scoping review method. With this, in addition to identifying what has been achieved until now, we discuss future research opportunities.

## 2. Materials and Methods

Scoping reviews are systematic bibliographical search processes whose results help investigate the knowledge and evidence about a specific topic as well as answer broad research questions since this methodology lets us know the theories, fundamental concepts, and knowledge gaps. In this study, we conducted a review by scope according to the guidelines of the PRISMA-ScR statement [11]. The latter aspect is the most relevant for our research purposes because the topic of applying innovative DA methods like ML tools to dairy farms is recent. Therefore, scoping reviews can detect knowledge gaps in a topic through the selected bibliographic review methodology. In this way, our research domain is adequate for performing a scoping review because studies regarding the application of ML tools to optimize herd management performance are relatively new or at least little explored. Interest in applying these tools in this field has also quickly increased over the past few years. To achieve our goal, we followed a standard scoping study procedure comprising five steps: (1) identify the research questions; (2) identify relevant studies; (3) select the criteria; (4) chart the data; and (5) collate, summarize, and report the results.

### 2.1. Research Questions

Following the methodology of a scoping review, to start the research and focus on the search, a broad research question is posed that aims to be the basis of our study. This general question is: How have DA tools been applied to dairy farms? However, given the amplitude of the subject and the comprehensive sources of the reports, we break down the main research question into four more specific research questions:

RQ1. What decision-making scope and level areas in dairy farms have been studied using DA with the objective of supporting future decision-making?

RQ2. Which has been the main focus of the dairy decision support literature regarding analytics (descriptive, predictive, or prescriptive), data (real-time or historical), and treatment of uncertainty (deterministic or stochastic)?

RQ3. What are the main ML and statistical methodologies used in the dairy farm decision support literature?

RQ4. Which software, programming languages, and data visualization tools were used in these studies?

To answer these questions, we developed a rigorously structured and sufficiently documented method to provide robust evidence and arguments.

### 2.2. Identify Relevant Studies

In our scoping review, we searched the Web of Science and Scopus databases for all relevant studies using keywords that we consider the best representatives of the study objective. The words used were dairy, milk, analytics, AI, big data, and neural networks. These words were entered into the search engines, considering the combinations of the Boolean operators available in the search engines that would allow us to obtain a small number of studies whose results were effectively related to the objective of this study. The specific keywords entered in both search engines are (dairy OR milk) AND (analytics OR AI OR big data OR neural networks). After we have obtained the results in both search engines, the available filters mentioned below are applied: (i) Article and review for document type; (ii) English for the language; (iii) for Scopus, the subject area entries are agriculture, veterinary, computer science, environmental science, decision sciences, and multidisciplinary; and for WOS, the research areas considered are agriculture and dairy animal science, veterinary sciences, food technologies, multidisciplinary sciences, agronomy, environmental engineering, agricultural economics policy, multidisciplinary applications, and informatics.

### 2.3. Selection Criteria

The definition of different inclusion and exclusion criteria was post-hoc because the researchers’ familiarity with the studies increased. In the first exclusion process (screening), we considered only articles published in peer-reviewed journals. We removed duplicate publications from the portfolio, reducing the number of articles. In the next exclusion step, the paper titles and keywords were individually verified to determine their alignment with the research topic (for example, the paper entitled “A comparative study of reproductive performance in organic and conventional dairy husbandry” was removed since the comparison did not involve DA methods or a decision-making problem). The remaining articles that passed the screening were then checked for abstracts’ alignment with the research topic on how DA tools have been applied to herd performance prediction on dairy farms. For example, considering keywords and abstract, the article “Predicting cow milk quality traits from routinely available milk spectra using statistical ML methods” was removed because the focus was on the prediction of milk quality traits, losing the herd performance perspective. Subsequently, the full texts of the selected results were reviewed. Those that were not consistent with the study objectives (e.g., the study “Improving farm decisions: the application of data engineering techniques to manage data streams from contemporary dairy operations” focuses on the integration and management of data on dairy farms, not on specific AI or DA methods to predict milk production or detect diseases) or papers with the text body not written in English were excluded.

### 2.4. Bibliometric Software

To perform the bibliometric study and analyze the selected articles, we used specialized software like Bibliometrix 4.0 [12] and Vosviewer 1.6.20 [13].

## 3. Results

### 3.1. Identify Relevant Studies and Selection Criteria

The result of our selection process of relevant articles is summarized in Figure 1. Of the 179 results, 151 are articles, and 28 are reviews. The scientific articles found are included in Appendix A, and the selected reviews are shown in Appendix B.

### 3.2. Chart the Data

Preliminary results allowed us to highlight that the selected articles have an average age of 5.95 years from their year of publication, indicating that the topic we are reviewing is relatively recent or has many more recent publications. There were 70 journals containing 151 papers, averaging around 2.2 publications per journal, indicating that the sources of our subject are diverse. The average number of authors per article is 3.56, and the number of articles that were made by a single author is two, which is a small number with regard to all the selected papers.

When reviewing the selected articles’ keywords, the words highlighted the most were related to milk production (dairy cow, 14 times; mastitis, 12 times; dairy cattle, 11 times; milk yield, 11 times) and to AI prediction techniques (neural network, 45 times; ML 22, times; deep learning, 17 times; prediction, 11 times; computer vision, 10 times). Words grouped in trigrams from words in the titles are also reviewed, in which the following groups stand out: artificial neural networks (25 times), day milk yield (nine times), multiple linear regression (seven times), ML algorithms (six times), convolutional neural networks (six times), among others. The aforementioned groups of words help solve the problem of milk production forecasting using ML methodologies, which also reflects the effort to make predictions using statistical methodologies (linear regression) in order to establish comparisons between the tools of ML and those of traditional methods.

Figure 2 indicates the relationship between the authors’ keywords in the 151 selected articles. This figure shows that the strongest relationship is generated by the Keyword Neural Networks, this concept being the one that acts as a general link of the other keywords.

In Figure 3, it is noteworthy that from 1994 to 2017, the number of publications related to milk production prediction using ML tools was relatively scarce and did not exceed four annual publications in this period. However, from 2018 onwards, there has been a significant growth in the number of studies per year on this topic, implying increased interest among researchers in exploring ML methodologies in dairy production.

The journals where the selected works were mainly published include Computers and Electronics in Agriculture (*n* = 37), Journals of Dairy Science (*n* = 18), and Animals (*n* = 7), which contain approximately 41% of total publications. The Indian Journal of Animal Science has six articles, and there were three publications each in the journals Sensors, Transactions of the American Society of Agricultural Engineers, and Livestock Science, and three apiece in the Canadian Journal of Animal Science. There are 57 journals that contain only one publication of the selected papers.

The author with the highest number of articles on the topic of milk production forecasting using AI tools is Lacroix R with 11 articles, followed by Wade K (*n =* 7), Cabrera (*n =* 5), Wrzesiak (*n =* 5), and Zhang (*n =* 5).

### 3.3. Management Aspects in Dairy Farms Connected with Decision-Making Problems

In an organization, there are three levels of decision-making: strategic, tactical, and operational. The strategic level focuses on long-term goals, overall direction, and resource allocation. Tactical decisions deal with implementing short- and medium-term strategies, with a view of months or years, and involve the allocation of specific resources. Finally, operational decisions concentrate on daily actions to ensure effectiveness and efficiency in routine activities, with a daily or weekly time horizon. These levels combine to achieve organizational objectives, ensuring coherence between strategy, tactics, and operational execution. When reviewing the selected articles, it is possible to notice that the scope of decision-making is mostly focused on directly studying milk yield estimation (as, for example, in the case of a time series where the kilograms of milk in a period of time are dependent on the production of milk in past periods) with 29% of the total studies, while 13% of all studies are dedicated to developing tools for strategic level decision-making, and 15% are at the tactical level. The second most studied area is the early detection of lameness and other diseases (not including mastitis) at 26%, and the case of the timely detection of mastitis at 13%. Both areas are dedicated to developing tools that support operational-level decision-making. It is important to note that the case of mastitis has been separated from other diseases since mastitis is an important disease that implies a greater cost in dairy farms. The fourth place in importance is occupied by reproductive measurements and diseases of calving, with 6% of the selected publications, where, in this case, it is roughly distributed equally at the strategic, tactical, and operational levels. The above results are summarized in Table 1, where cases with low numbers of publications, such as detecting animal behaviors with cameras, detecting animals’ social behavior, and calving day prediction, are omitted.

### 3.4. Data Analytics

According to the type of analytics used in the selected works, we can note that 87% of the publications focused on predictive models, which is in line with the importance of milk production and the number of studies on milk yield prediction. The cases that use descriptive and prescriptive analytics are less prevalent compared to the aforementioned predictive cases. Table 2 indicates that the data used in the investigations are mostly historical data, with 70% of the case studies, compared to the cases that use real-time data, which are used in 25% of the cases, even though none of the papers developing predictive models validated them on a different farm. This poses the question of the need for calibrating model parameters for individual farms and the validity requirements of the model in different herds.

### 3.5. Coverage of Machine Learning and Statistical Methods by Revised Papers

Regarding ML methodologies, Table 3 indicates that the methods of artificial neural networks and convolutional neural networks dominate with 47% and 24%, respectively, with the random forest method in third place at 12% of cases. A total of 13% of cases do not apply this type of tool but focus, for example, on methods to efficiently obtain large amounts of time suitable for optimization in decision-making on farms: big data, IoT, sensors, etc. Artificial neural networks have been widely considered prediction methods because they deliver greater precision in forecasts than other ML methods. Many of the selected works, along with ML methodologies, use statistical methods in order to make comparisons between the two types of methodologies. As Table 3 indicates, of the 151 articles selected, 56 include traditional methods like linear regression and simulation. Considering only the publications that use statistical methods, the most used method is linear regression, with 50% of cases, while other methods, such as simulation and logistic regression, are scarcely used, each with 4% of cases.

Regarding the evaluation of the statistical models’ parameters, 28 studies report the determination coefficients (R2) at an average of 76%, with a range between 16% and 97%. Of the 28 studies that calculate the coefficient of determination, 79% of the cases develop models for the prediction of milk production. When considering the accuracy results of sixteen models using artificial neural networks, they averaged 76% (minimum 23% and maximum 99%); recall reported only in five papers averaged 75% (minimum 64% and maximum 97%); precision reported in four papers averaged 75% (minimum 50% and maximum 97%); and F1-Score metrics reported in only two papers, with 76% and 99.9%.

Table 4 shows that the backpropagation algorithm for artificial neural network models is the one used in 62% of cases, while 38% did not indicate what type of algorithm was used. In addition, neurons in artificial neural network models preferentially use sigmoid hyperbolic tangent activation functions in 34% of cases, and 57% of studies employing artificial neural networks do not mention what type of activation function perception was used. Finally, we can note that the input layer in proposed artificial neural networks considers a number of three to six independent variables as inputs and, in most cases, a sole dependent variable as output. Our review also shows that artificial neural network models employ a small number of hidden layers; the majority of models use one or two hidden layers most frequently. A total of 96% of all papers using one or two hidden layers were published between 2003 and 2023, and 62% of papers were published between 2013 and 2023.

### 3.6. Treatment of Uncertainty

The consideration of uncertainty is not very common in the 151 selected articles. The vast majority, that is, 117 papers (77% of the total studies), use deterministic models; few studies cover uncertainty by considering stochastic models, at only 5% of the selected articles (see Table 5).

### 3.7. Reported Software for Data Analytics

A quarter of the selected articles do not mention the statistical software used to perform the statistical analysis. This is explained by the fact that they use other general programming languages, like Python, which have specific libraries for this kind of analysis. The most used DA tools are Matlab, with 20% of cases, followed by Python and R, with 19% and 15% respectively (Table 6). A small number of publications mention the libraries used for developing the ML methods, including Keras (5%), Pytorch (2%), and YOLO (4%), in the case that Python is used. In addition, most of the software used belongs to the category of modeling language, which implies that the same tool presents a graphical environment and can be used for data visualization.

## 4. Discussion

### 4.1. General Overview of the Data

The main objective of our research work is embodied in the general research question that asks about the way data analytics tools have been applied to dairy farms. From our findings (Figure 3), this topic has been appealing to the research community in recent times; since 2018, there has been a considerable increase in the number of publications dealing with the topic of data analytics in dairy farms. This observation is reinforced by the fact that the average publication age of the 151 selected articles is only approximately six years. In agreement with us, Shine and Murphy [3] carried out a review of the application of ML in dairy farms where they mention that 74% of the articles selected were dated after 2017, while Slob et al. [9], in a similar review of the application of ML to dairy farm management identified that in 2018 there was an increase in the use of several ML algorithms such as decision trees (one article in 2017, four in 2018 and six in 2019) and neural networks (two articles in 2017, two in 2018, and six in 2019). The coincidence in the growth of scientific publications from 2018 in the fields of ML and DA is due to the synergies and close relationship between these two disciplines. In particular, the development of ML techniques can significantly enhance the ability to analyze and derive valuable insights from complex data sets, such as those that may be involved in milk production. For instance, ML makes it possible to identify diseases, like mastitis in cows, faster through somatic cell counting or monitoring the behavior of animals by computer vision to improve animal welfare. Another trend observed in recent years regarding DA is the addition of Big DA to reinforce the connection with digital data. In this sense, Lokhorst et al. [14] reviewed big data studies for dairy farms and reported that the number of articles on this topic had increased since 2007. The difference between the year in which the number of studies begins its rapid growth for data analytics in 2018 and that of big data in 2007 is due to the development of sensor devices and the automatic gathering of large amounts of data prior to developing and using methods for handling and analyzing it. In general, it is likely that the increased interest that has arisen recently in the topic of data analytics in dairy farms is due to the need to remain competitive, sustainable, eco-friendly, and more economically efficient in producing milk. Social pressure and change in attitudes toward CO2 footprint and animal products may also affect the product, although it is not directly considered in the revised papers.

Figure 2 provides insight into the keywords used in selected papers and indicates that the predominant words can be classified into two thematic groups, one related to the object of study (milk yield, dairy cattle, mastitis, and dairy cows) and another considers the techniques used to carry out those studies (ML, deep learning, prediction, computer vision, neural network and convolutional neural network), which was coherent with the title word clouds via trigrams (calculated results but not displayed). All the preceding concepts are strongly related to the purpose of dairy farm production forecasting and point out the relevant problems, such as timely identification of costly diseases, such as mastitis or lameness, all of this through the use of ML techniques, including artificial neural networks and convolutional neural networks. Sometimes, traditional statistical methods are confronted with new ML methods. For example, Dongre et al. [15] and Manoj et al. [16] compared the efficiency of artificial neural networks and multiple linear regression analysis for prediction of first lactation 305-day milk yield and concluded that the level of accuracy of the neural network models presented in this study is higher than that obtained by the multiple linear regression method. Edriss et al. [17] make a similar comparison between neural networks and linear regression, considering the performance of the second parity of animals, and they also conclude that neural networks perform better in forecasting. The authors ([17]) justify such differences in the quality of predictions since artificial neural networks have the ability to model nonlinear relationships and capture more complex patterns [17].

Neural networks (see Figure 2) interact strongly with other important concepts like mastitis and dairy production. Artificial neural network methods have multiple applications in dairy farms, such as early detection of lameness and mastitis, prediction of milk production, or detection of cows with artificial insemination problems [8]. Other keywords that generate minor interaction nuclei with other ideas are deep learning, computer vision, mastitis, milk yield, etc. There is also an isolated group of keywords dominated by the words DA, cloud computing, fog computing, and the Internet of Things (IoT). These keywords reflect researchers’ efforts to generate good computer methods and tools for data acquisition and efficient hosting in dairy farms. This way, it is possible to perform the analyses automatically on the farm and in a timely manner, processing data into valuable information for better decision-making [7,14].

### 4.2. Management Aspects in Dairy Farms Connected with Decision-Making Problems

The answer to the first research question (RQ1) is covered by the information presented in Table 1. Hence, the most important decision-making areas of the selected publications are estimating milk production (29%), the early detection of lameness and other diseases (excluding mastitis) (26%), detection of mastitis (13%), reproductive measurements, and diseases of calving (6%). Within milk production estimation, the dominating decision-making scope is at the tactical and strategic levels. This is due to variables affecting the production, like the long-term decisions related to the size of the herd, investment in new facilities, or the implementation of new technologies, which implies a long-term view of the company and involves the general management. For instance, Saha and Bhattacharyya [18] study the relationship between artificial insemination with statistical and ML tools; their results can be used for long-term decision-making by influencing the level of milk production by manipulating the genetic potential of animals.

Decisions involve the medium term, such as the hiring of the farm staff or the scheduling of milking processes. For example, Murphy et al. [19] studied neural network models that aimed to make milk production forecasts for 305-day cycles, but they also considered models that forecasted milk production for 10, 30, and 50 days, which could, therefore, be useful for medium-term decision-making.

Regarding lameness detection and other diseases and early detection of mastitis, the scope of these studies is concentrated on operational decision-making because these diseases have a direct impact on the daily activities of the dairy farm, and early detection is important for the timely separation of sick animals, adjustments in feeding or the application of veterinary treatments, and in that way, avoid losses in production and ensure product quality. In the case of early detection of diseases, it is necessary to develop sensors that provide reliable, real-time data so that farmers can act in a timely manner [2]. This implies retrieving information necessary for decision-making in the short term.

### 4.3. Data Analytics and Treatment of Uncertainty

As shown in Table 2, the type of analytics is predominantly predictive, and the data used are generally historical rather than real-time. Table 5 shows that in 19% of the selected articles, uncertainty was not relevant because they were more focused on technology development, like big data and cloud computing techniques, as was the case of Kulatunga et al. [20]. We answer the second research question (RQ2). The dominance of deterministic models over stochastic models may be due to the fact that deterministic models are simpler, based on mathematical relationships considering well-known causal relationships between variables, such as the relationship between feeding and milk production. As mentioned above, one of the most relevant aspects of this type of study is the prediction of milk production, which corresponds to the category of predictive models, tools that can support the estimation of future milk production, and with that, the planning of dairy herd operations, the estimation of the number of farm staff, and resources required. The preference for historical data may be because, for example, it is essential to know the patterns and trends of milk production. It may be relevant to study the application of stochastic models to generate more realistic models [21], consider prescriptive models in order to expand automation in farms and explore the application of data generated in real time that, in the short term, can be used in DSS, which generate suggestions at appropriate times in order to optimize operations on farms. In Vidal [21], it is mentioned that there are fields of operations research where there is currently a greater interest in stochastic models over deterministic models due to their advantages; however, the use of deterministic models is maintained. Precision livestock farming is defined as “real-time monitoring technologies aimed at managing the smallest manageable production unit, also known as the ‘sensor-based’ individual animal approach” [22], and it can contribute to the improvement in a well-managed system by increasing the information available in the early detection of diseases or lameness, food consumption, and eating behavior, quantifying the pain and stress of animals, heart rate detection, body condition score, etc.

### 4.4. Machine Learning and Statistical Methods

Regarding the third research question (RQ3), looking at Table 3, the ML methods mostly used are artificial neural networks and convolutional neural networks. Shine and Murphy [3] mention that the most commonly used ML algorithms in the study of dairy farms are those based on decision trees (54%) and followed by those of neural networks (50%, considering artificial neural networks and convolutional neural networks together). Our work coincides with Shine and Murphy [3] on only 16 selected articles, which corresponds to 11% of our 151 selected articles. This difference may be due to the fact that different search engines were used (Scopus, Science Direct, IEEE, Google Scholar, and MDPI), different keywords, selection criteria, and only the articles available until the first half of 2021.

In artificial neural network models, backpropagation algorithms are used, and they consider mostly activation functions of their sigmoid hyperbolic tangent perceptron. The number of input variables in these models mainly lies between three and six independent variables, while in most cases, one output variable (dependent variable) is used. The number of hidden layers used is small, with the highest frequency being for one or two hidden layers. This result agrees with Perdigón-Llanes and González-Benítez [23], who review the application of neural networks in the prediction of milk production, mentioning that most of the articles selected by their study use two hidden layers. When considering the descriptive statistics for the accuracy of models using artificial neural networks (average 76%, standard deviation 20%, minimum 23%, and maximum 99%), results suggested that some models are highly accurate while others have much lower performance. Descriptive statistics for recall (average 75%, standard deviation 13%, minimum 64%, and maximum 97%) indicated that some models are better at generating items correctly identified as positive out of the total positives, precision (average 75%, standard deviation 22%, minimum 50%, and maximum 97%), which notes that models’ accuracy varies widely and with variability suggesting that some models may have a high number of false positives, and F1-Score metrics (average 88%, standard deviation 17%, minimum 76%, and maximum 99.9%), indicating that some models achieve a significant balance between accuracy and recall, while others may emphasize one of these metrics. All these results show us that artificial neural network models do not all generate results of the same quality level. Some show a strong track record in making correct predictions, while others fall short. The reason for this difference could be because of different data, different types of neural networks, or training method variations across the studies.

Several studies indicate the importance of DSS development in dairy farms. These support systems can be greatly improved by including more ML tools. For example, Balhara et al. [24] mentioned that Decision Support Systems (DSS) are integral in dairy farms, utilizing computer-based models and data analysis for informed decision-making. These systems, often data-driven, optimize resource utilization, enhancing productivity and economic outcomes in livestock production. The adoption of commercially available DSS reflects their vital role in modern dairy management, offering a professional approach to decision support with a focus on scientific advancements.

According to the results presented, the neural network method is the most widely used when considering ML methodology application. The present review also indicates that simulation models are scarcely used. These characteristics suggest that combining both methods to generate a DSS in future research may be beneficial. Neural network methods that generate considerable adjustments could also be used together with simulation models in order to have a slightly more realistic reproduction of some important aspects that occur in this type of environment. Von Rueden et al. [25] mention that some applications can benefit from the combination of simulation and ML techniques, whether the simulation method assists ML or ML supports simulation.

### 4.5. Software for Data Analytics

Related to the fourth research question (RQ4), Table 6 shows that the most used software is MATLAB, but a decline has been observed in recent years, while the preferred programming languages are MATLAB, R, and Python. Python has been predominant in recent papers, and this may be due to the level of accessibility and libraries required to develop AI-based models. It is efficient in the use of large amounts of data and is very useful and flexible for data exploration and visualization. Most data visualization is performed with the same programming languages, like Matlab, Python, and R.

### 4.6. Gaps in the Literature and Future Outlook

With regard to the third research question posed in our work (What are the main ML and statistical methodologies used in the dairy farms decision support literature?), we have identified most research gaps in applying DA tools on dairy farms:There is a need to explore more accurate models, like stochastic models, for predicting milk production. Stochastic models consider the intrinsic randomness that a system can have, and currently, in some fields, they are of greater interest to researchers than deterministic models [21];The literature does not provide references to dairy farms about the development and use of prescriptive analysis. While the use of descriptive analytics is rather common, the use of predictive analytics is lower. Lepenioti et al. [26] indicate that prescriptive analytics is currently less developed than descriptive and predictive analytics, considering its development as the next step toward increasing DA maturity;The combination of different methods developing synergies is another interesting research line claimed by different authors like von Rueden et al. [25]. They remark on the complementarity of simulation and ML. When reviewing the level of hybridization between AI and simulation tools in dairy farms, we did not find studies dedicated to this combination of methods. To explore ways in which ML tools are combined with simulation methods, we can review possible applications of these tools in other non-livestock species, even in other situations a little more distant from livestock where DSS is presented with the implementation of the two tools, for example, in the case of the development of autonomous vehicles [27]. These approaches are near the development of digital twins and concepts of augmented reality or virtual reality, seeking realistic simulation environments [28];Literature reports on studies that use mostly historical data. However, with the greater development of sensors, the use of IoT, cloud computing, fog computing, and big data, real-time data can acquire greater importance in the models used. The increased use of AI tools in DSS can greatly improve the adoption of these systems.

Our review uses only the research published in the WOS and Scopus databases as the most relevant in the scientific context. It is unlikely that other important works published in other databases have been excluded from our analysis. Works related to our topic and published in the gray literature have also been excluded because not having peer review would add bias to our analyses. We consider only studies in English because it is the predominant language in the scientific literature, which guarantees access to high-quality research and facilitates the review and comparison of studies; however, this may result in us not considering important articles not yet published in this language. The selection process based on search terms such as “milk” and “neural networks” may have introduced bias into the results. By focusing on these terms, studies that address the prediction of milk production and the use of neural networks may have been favored, perhaps excluding research that uses other AI approaches or that deals with different aspects of dairy farms.

## 5. Conclusions

We have concentrated on reviewing the articles investigating the use of DA methodologies in dairy farms. The world population has grown rapidly, which will lead to increased bovine milk demand. In turn, the necessary production rises for this demand will only be achievable by applying technology for timely animal disease detection, selecting animals with desirable genetic characteristics, and optimizing food and water delivery.

To solve our research questions, we followed a scoping review methodology and selected 151 research articles. These investigations focus on obtaining good predictions of milk production and early disease detection, highlighting mainly mastitis and lameness. These animal health problems are determinants for milk production forecasting. Another important area of study is reproduction. The studies are dedicated to obtaining decision-making support tools at a strategic level in the case of determining milk production, while in the detection of lameness and other diseases and the case of mastitis, the support for decision-making is at the operational level. We have found that interest in the subject is recent; predictive analytics predominate; the models used to make the predictions are deterministic, and the data types used in the models are 70% historical data and 25% real-time data. The tools of DA preferred by researchers in their work are Matlab, R, and Python. Most of the programming languages used were modeling languages, making the same tools useful for data visualization. In ML methodologies, the most used methods are artificial neural networks and convolutional neural networks. In the case of statistical methodologies, the most used method is multiple linear regression. The studies show that ML methods have greater predictive capacity than statistical methods. Simulation tools are rarely used to study bovine milk production. The articles do not show the combination of simulation tools and ML, and this combination may present a possibility for future dairy production research.

## Figures and Tables

**Figure 1 animals-15-01291-f001:**
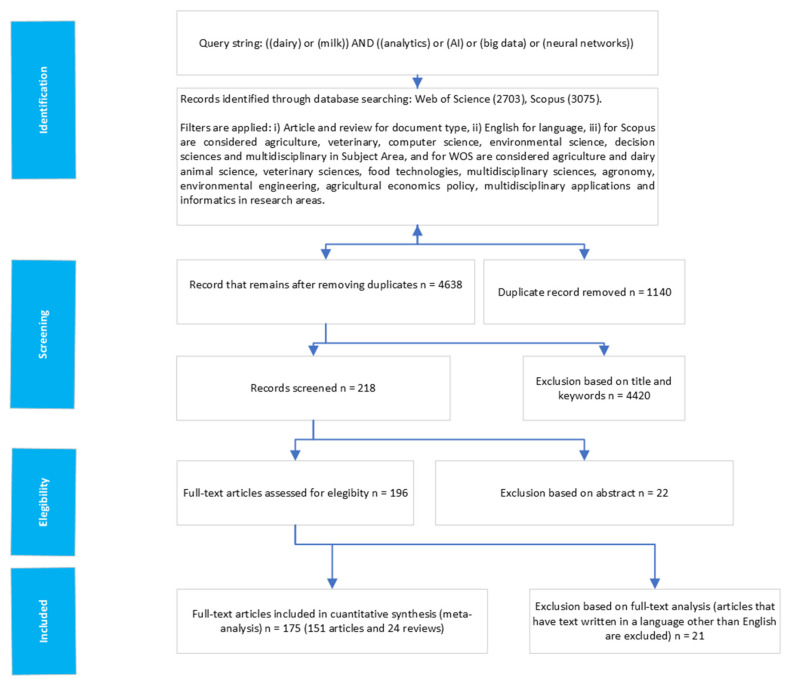
Summary of the article selection process. Prepared by the author using Microsoft Visio 2024 (Standard).

**Figure 2 animals-15-01291-f002:**
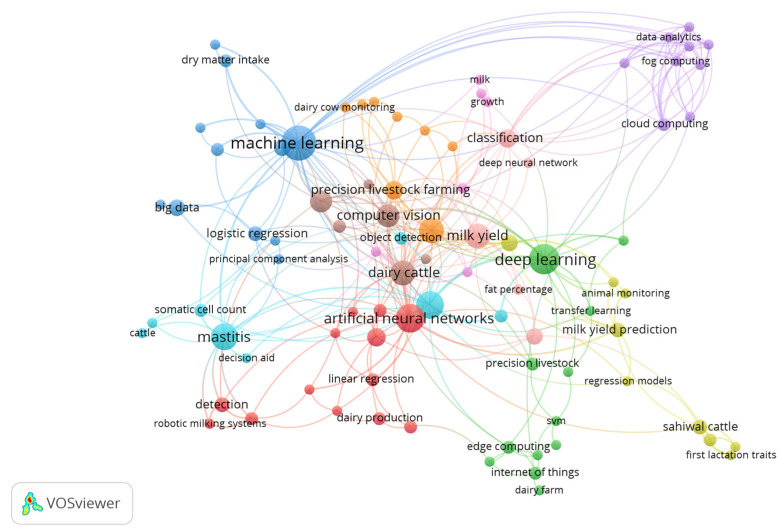
Relationship between author keywords. Prepared by the author using Bibliometrix software.

**Figure 3 animals-15-01291-f003:**
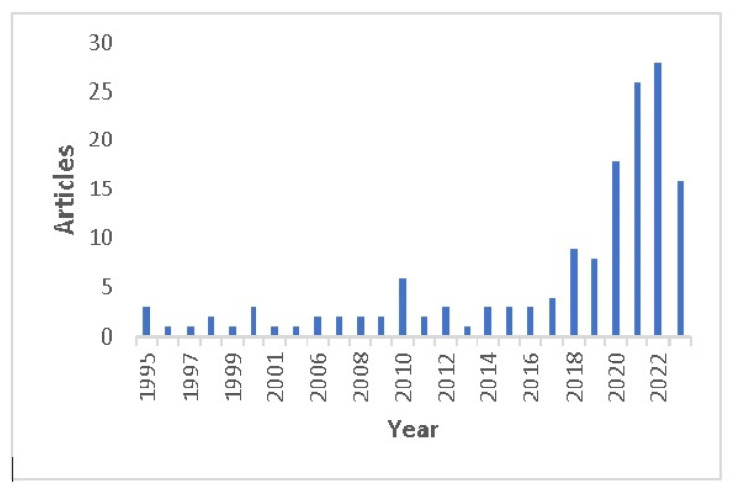
Annual scientific production of the selected articles in our studio.

**Table 1 animals-15-01291-t001:** Details of decision-making scope in each of the selected studies, as well as the strategic, tactical, or operational levels of decision-making.

		Decision-Making Level		
		Strategic	Tactical	Operative
Scope of decision-making	Milk yield estimation	13% (A8, A19, A20, A28, A41, A43, A45, A50, A55, A58, A70, A74, A76, A78, A80, A81, A84, A93, A125,A144)	15% (A3, A5, A9, A30, A31, A39, A46, A48, A57, A59, A73, A79, A82, A83, A109, A111, A113, A118, A120, A138, A145, A150)	1% (A52, A142)
	Early detection of lameness and other diseases	2% (A26, A54, A114)	1% (A124)	23% (A2, A14, A23, A24, A25, A47, A56, A63, A72, A75, A88, A91, A94, A106, A110, A112, A115, A116, A117, A119, A121, A126, A128, A129, A130, A131, A132, A133, A135, A136, A139, A140, A147, A151)
	Mastitis detection	1% (A1)	1% (A4)	11% (A6, A11, A13, A21, A29, A33, A42, A85, A92, A95, A96, A98, A99, A100, A101, A104, A127)
	Reproductive measurements and calving diseases	1% (A22, A65)	3% (A12, A17, A27, A152)	2% (A18, A34, A64)
	Food intake	1% (A7)	3% (A35, A51, A53, A67, A87)	1% (A103)
	Not applicable	4% (A10, A15, A32, A44, A77, A90, A105, A107, A122, A123, A137, A148, A149, A146)	1% (A108, A141)	1% (A97, A143)
	Body weight and physiology	1% (A61)	1% (A37, A66)	1% (A36)

**Table 2 animals-15-01291-t002:** Frequency and percentage of different characteristics in the selected works: type of analysis and type of data.

	Type (Percent)	Papers
Type of analytics	Predictive (87%)	A1, A2, A3, A4, A5, A6, A7, A8, A9, A11, A12, A13, A14, A17, A18, A19, A20, A21, A22, A23, A24, A25, A27, A28, A29, A30, A31, A33, A34, A35, A36, A37, A38, A39, A40, A41, A42, A43, A45, A46, A47, A48, A50, A51, A52, A53, A54, A55, A56, A57, A58, A59, A61, A64, A65, A66, A67, A68, A69, A70, A71, A72, A73, A74, A75, A76, A78, A79, A80, A81, A82, A83, A84, A85, A86, A87, A88, A89, A91, A92, A93, A94, A95, A96, A97, A98, A99, A100, A101, A102, A103, A104, A106, A107, A108, A109, A110, A111, A112, A113, A114, A115, A116, A117, A118, A119, A121, A123, A124, A125, A126, A127, A128, A129, A130, A131, A132, A133, A135, A136, A138, A139, A140, A143, A144, A145, A146, A147, A150, A151, A152
	Descriptive (5%)	A120, A134, A141, A142, A148, A149
	Prescriptive (3%)	A60, A62, A63
	Not applicable (7%)	A10, A15, A16, A26, A32, A44, A77, A90, A105, A122, A137
Input Data types	Real-time (25%)	A15, A16, A18, A22, A23, A25, A26, A29, A60, A61, A62, A63, A64, A69, A87, A89, A102, A103, A110, A112, A116, A117, A119, A121, A127, A128, A129, A130, A131, A132, A133, A135, A136, A140, A147, A151
	Historical (70%)	A1, A2, A3, A4, A5, A6, A7, A9, A10, A11, A12, A13, A14, A17, A19, A20, A21, A24, A27, A28, A30, A31, A33, A34, A35, A36, A37, A38, A39, A40, A41, A42, A43, A45, A46, A47, A48, A50, A51, A52, A53, A54, A55, A56, A57, A58, A59, A65, A66, A67, A70, A71, A72, A73, A74, A75, A76, A78, A79, A80, A81, A82, A83, A84, A85, A86, A88, A91, A92, A93, A94, A95, A96, A97, A98, A99, A100, A101, A104, A106, A107, A108, A109, A111, A113, A114, A115, A118, A123, A124, A125, A126, A134, A138, A139, A141, A142, A143, A144, A145, A146, A148, A149, A150, A152
	Third Party Sources (1%)	A8
	Not applicable (6%)	A32, A44, A68, A77, A90, A105, A120, A122, A137

**Table 3 animals-15-01291-t003:** Frequency and percentage of machine learning and statistical methodologies used in the selected works.

	Methodology	Percentage	Papers
Machine Learning Methodology	Artificial Neural Network	47%	A1, A2, A3, A4, A6, A7, A9, A11, A12, A13, A14, A17, A19, A20, A21, A27, A28, A30, A31, A33, A34, A35, A36, A37, A38, A39, A40, A42, A43, A45, A46, A47, A48, A50, A53, A56, A57, A58, A59, A60, A64, A66, A67, A68, A72, A73, A74, A76, A78, A79, A80, A81, A82, A83, A84, A85, A88, A94, A104, A109, A111, A113, A115, A118, A120, A124, A132, A142, A143, A145, A152
	Convolutional neural network	24%	A2, A18, A23, A24, A25, A41, A51, A69, A71, A75, A87, A89, A96, A102, A103, A106, A110, A112, A116, A117, A119, A121, A127, A128, A129, A130, A131, A133, A135, A136, A139, A140, A142, A147, A151
	Not applicable	13%	A10, A15, A16, A29, A32, A44, A52, A54, A55, A70, A77, A90, A101, A108, A113, A120, A126, A138, A139, A152
	Random forest	12%	A2, A26, A65, A82, A86, A91, A92, A94, A99, A100, A101, A104, A108, A113, A126, A139, A142, A152
	Unspecified machine learning methods	3%	A61, A62, A63, A65,A152
	SVM	7%	A2, A8, A67, A86, A104, A106, A114, A120, A126, A139, A142
	Decision trees	9%	A86, A91, A96, A97, A98, A100, A104, A106, A108, A114, A120, A142, A144, A152
	Fuzzy logic	1%	A47, A93
	K-nearest neighbors	5%	A14, A26, A106, A108, A142, A144, A152
	k-means	1%	A22
Statistical methodology	Linear regression	50%	A3, A5, A8, A17, A31, A39, A46, A50, A51, A53, A54, A55, A57, A58, A59, A61, A70, A73, A76, A78, A80, A83, A97, A108, A111, A124, A125, A150
	Simulation	4%	A29, A107
	Linear discriminant analysis	4%	A4, A9
	Logistic regression	4%	A27, A94
	Wood model	4%	A74, A81
	Other	21%	A2, A7, A12, A30, A38, A40, A52, A61, A85, A108, A118, A146

**Table 4 animals-15-01291-t004:** Frequency and percentage of algorithms in artificial neural networks and the activation functions in the selected works that use neural networks.

		Percent	Paper
Algorithm used in artificial neural networks	Back-propagation	62%	A1, A3, A4, A5, A6, A11, A12, A14, A19, A21, A23, A27, A28, A30, A31, A36, A37, A39, A40, A41, A42, A43, A44, A46, A47, A48, A50, A56, A57, A58, A59, A67, A72, A73, A74, A76, A79, A80, A81, A84, A86, A88, A104, A111, A115, A145
	Not mentioned	38%	A2, A7, A9, A13, A17, A20, A33, A34, A35, A38, A45, A53, A60, A64, A66, A68, A82, A83, A85, A109, A113, A118, A120, A124, A132, A142, A143, A152
Activation function	Hyperbolic tangent	34%	A1, A3, A5, A6, A9, A37, A39, A43, A48, A50, A51, A53, A58, A59, A72, A73, A76, A79, A80, A83, A84, A86, A109, A113, A115
	RELU	1%	A20
	Sigmoid logarithm	4%	A5, A86, A124
	Lineal	5%	A50, A74, A86, A115
	Exponential	1%	A74
	Not specified	57%	A2, A7, A11, A12, A14, A17, A19, A21, A27, A28, A31, A33, A34, A35, A36, A38, A40, A42, A45, A46, A47, A56, A57, A60, A64, A66, A68, A78, A81, A82, A85, A88, A94, A95, A111, A118, A120, A132, A142, A143, A145, A152

**Table 5 animals-15-01291-t005:** Frequency and percentage of different characteristics of the selected works as a treatment of variability, type of analysis, and type of data.

		Percent	Papers
Treatment of variability	Deterministic	77%	A1, A2, A3, A4, A5, A6, A7, A8, A9, A11, A12, A13, A14, A17, A18, A19, A20, A21, A22, A23, A24, A25, A27, A28, A30, A31, A33, A34, A46, A47, A48, A50, A51, A53, A54, A55, A56, A57, A58, A60, A61, A62, A63, A65, A66, A67, A68, A69, A70, A71, A72, A73, A74, A75, A76, A78, A79, A80, A81, A82, A83, A84, A85, A86, A87, A88, A89, A91, A92, A94, A95, A96, A97, A98, A100, A102, A103, A104, A106, A108, A109, A110, A111, A112, A113, A114, A115, A116, A117, A118, A119, A121, A123, A124, A125, A126, A127, A128, A129, A130, A131, A132, A133, A135, A136, A138, A139, A140, A142, A143, A144, A145, A147, A150, A151, A152
	Stochastic	5%	A29, A52, A93, A99, A103, A104, A107, A146
	Not applicable	19%	A10, A15, A16, A26, A32, A35, A36, A37, A38, A39, A40, A41, A42, A43, A44, A45, A59, A64, A77, A90, A101, A105, A120, A122, A134, A137, A141, A148,149

**Table 6 animals-15-01291-t006:** Frequency and percentage of the data analytics tools of the selected works and the basic or modeling language type (X means present).

Data Analytics and Machine Learning Tools	Percentage	Basic Language	Modeling Language	Papers
Not mentioned	44%			A1, A5, A10, A11, A15, A16, A22, A23, A26, A29, A32, A35, A38, A41, A44, A54, A63, A65, A68, A71, A74, A77, A78, A84, A85, A88, A89, A90, A96, A97, A103, A105, A106, A109, A112, A114, A116, A120, A121, A122, A125, A127, A128, A129, A131, A134, A137, A138, A139, A140, A142, A144, A145, A146, A148, A149, A151
Matlab	20%		X	A3, A6, A17, A19, A30, A31, A36, A39, A40, A47, A57, A58, A59, A62, A69, A70, A73, A76, A79, A80, A83, A86, A93, A111, A124, A133
R	15%		X	A2, A7, A8, A34, A50, A51, A52, A53, A56, A61, A67, A72, A82, A91, A92, A94, A99, A118, A126
Python	19%		X	A18, A20, A24, A25, A64, A66, A75, A87, A95, A102, A104, A107, A108, A110, A117, A119, A130, A132, A135, A136, A143, A147, A152
Statistica	5%		X	A12, A27, A46, A48, A55, A81
Neuralware	5%		X	A28, A37, A42, A43, A45, A112
SAS	5%		X	A34, A39, A81, A92, A123, A150
SPSS	4%		X	A9, A33, A40, A55, A141
TensorFlow	3%		X	A14, A20, A24, A95
C++	0.8%	X		A72
H_2_O	2%		X	A53, A61, A113
MES (Model Evaluation System)	2%		X	A51, A53
Neural Works Profesional II	0.8%		X	A21
Weka	2%		X	A92, A98, A101
Force 2.0	0.8%		X	A83
Neucube	0.8%		X	A60
Neuroshell	0.8%		X	A4
Java	0%	X		
Viscovery	0.8%		X	A13
SOMine	0.8%		X	A13
Aiyude Neurointelligence	0.8%			A13

## Data Availability

The original contributions presented in this study are included in the article. Further inquiries can be directed to the corresponding author.

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
