# Peer review of "AI and Data Analytics in the Dairy Farms: A Scoping Review"

_animals, 2025, doi:10.3390/ani15091291_

Round 1

Reviewer 1 Report

Comments and Suggestions for Authors

The authors use the phrase "dairy industry" too loosely throughout the manuscript, which may suggest that the review concerns the activities of the food industry, including dairies. Meanwhile, the work discusses issues taking place at the level of a dairy farm (animal breeding, housing, feeding; animal welfare; dairy cows performance, etc.). The problem of the incorrect use of the phrase "dairy industry" requires correction throughout the manuscript.

The readability/quality of figures should be improved (too low resolution/contrast)

The citations in the manuscript text should be corrected according to the References and the way of citing publications in the text should be corrected following the requirements of the Animals journal (for example in lines 269-271).

Lines 459-466: "...like:" suggests that there will be a bullet point in the text, but there is nothing like this; in turn, the paragraph ends with ". And" What is this about? A similar problem (unfinished sentence?) concerns lines 571-573.

The editorial style should be adjusted (e.g., subsections should be marked, table names should be above the tables and not below them).

Comments on the Quality of English Language

---

Author Response

The authors use the phrase "dairy industry" too loosely throughout the manuscript, which may suggest that the review concerns the activities of the food industry, including dairies. Meanwhile, the work discusses issues taking place at the level of a dairy farm (animal breeding, housing, feeding; animal welfare; dairy cows performance, etc.). The problem of the incorrect use of the phrase "dairy industry" requires correction throughout the manuscript.

The change of the term "dairy industry" to "dairy farms" in the title and throughout the text of the manuscript is justified to improve the precision and clarity of the study's approach. While "dairy industry" may suggest a broader scope that includes all activities related to the production, processing and marketing of dairy products, the term "dairy farms" refers specifically to activities and operations within dairy farms, such as breeding, housing, feeding, animal welfare and performance of dairy cows.

The readability/quality of figures should be improved (too low resolution/contrast)

Figures are corrected to improve resolution.

The citations in the manuscript text should be corrected according to the References and the way of citing publications in the text should be corrected following the requirements of the Animals journal (for example in lines 269-271).

All references are corrected to comply with the format of the journal Animals. For example, on line 86.

Lines 459-466: "...like:" suggests that there will be a bullet point in the text, but there is nothing like this; in turn, the paragraph ends with ". And" What is this about? A similar problem (unfinished sentence?) concerns lines 571-573.

The first part is corrected in line 499. The second part corrected on lines 620-648.

The editorial style should be adjusted (e.g., subsections should be marked, table names should be above the tables and not below them).

Corrects subsections and table headings.

Reviewer 2 Report

Comments and Suggestions for Authors

Dear authors,

I have reviewed your paper. The subject is interesting and as you state it is emerging and relevant. However, I think you should be more clear with the aim – to me there is a big difference between methods used in the industry and methods that are still on the research scale (and I interpret it as you are focusing on the latter).

Also, you could be more clear in the description of exclusion of studies. It is to me not clear why studies were selected or not (also not after getting the examples, which otherwise I think is a good way to present). I also would want a more thorough discussion on how your selection process could have affected the results (e.g. having milk and neural networks as search terms and finding that many/most studies covers milk production prediction and using neural networks).

Detailed remarks:

Line 99: none … do not…: Unclear what is meant here.

Selection criteria: It is to me not obvious that lameness prediction is unrelated to performance – I would need more guidance in how certain papers were selected and discarded.

Line 228: It would be more relevant to analyse key words and titles together, as key words typically are chosen to increase possibilities to find appropriate papers when performing a search, and all search engines also covers at least the title.

Line 314: Avoid using the term significantly as it may be confused with statistical significance.

Line 326-327: As Neural network was one of the search terms, it is not surprising that it is the most common method in the selected papers.

Line 355-356: Highest frequency of models?

Table 5: What is meant by machine learning (3%)?

Line 395: I would call I t Dairy research rather than dairy industry.

Line 429-430: Are all the concepts identified focused on the reason for forecasting/predicting production? I find it difficult to see the figure (low resolution) but I am not convinced that everything that is put down relates to why it could be good to predict production?

Line 528-539: The variability between published results will of course depend on input data quality, model validation methods, data set size and homogeneity and other factors. I am not sure what you want to say with this section. All the papers you have been using are published, perhaps that could also be discussed (is there a threshold on accuracy/R2/other measures to get published).

558-563: Unclear writing (e.g. “in cases where what can be useful when…”)

Author Response

I have reviewed your paper. The subject is interesting and as you state it is emerging and relevant. However, I think you should be more clear with the aim – to me there is a big difference between methods used in the industry and methods that are still on the research scale (and I interpret it as you are focusing on the latter).

It is answered on lines 101 to 112.

Also, you could be more clear in the description of exclusion of studies. It is to me not clear why studies were selected or not (also not after getting the examples, which otherwise I think is a good way to present).

It is answered on lines 214-219.

 I also would want a more thorough discussion on how your selection process could have affected the results (e.g. having milk and neural networks as search terms and finding that many/most studies covers milk production prediction and using neural networks).

Thanks for your comment. It is considered on lines 657-662.

 Detailed remarks:

Line 99: none … do not…: Unclear what is meant here.

It is answered on lines 101.

Selection criteria: It is to me not obvious that lameness prediction is unrelated to performance – I would need more guidance in how certain papers were selected and discarded.

It is corrected in lines 214-217. It is a mistake because the article mentioned on lameness is A94 and therefore it was considered.

Line 228: It would be more relevant to analyse key words and titles together, as key words typically are chosen to increase possibilities to find appropriate papers when performing a search, and all search engines also covers at least the title.

The requested joint review can be found on lines 251-258.

Line 314: Avoid using the term significantly as it may be confused with statistical significance.

Done on lines 337-338.

Line 326-327: As Neural network was one of the search terms, it is not surprising that it is the most common method in the selected papers.

Thank you for your observation. It is true that when including "neural network" as a search term, we find many studies that use this methodology. However, we decided to include it because neural networks are very relevant and useful in dairy production and likely connected with future methods. They have proven to be very effective at analyzing complex data and predicting outcomes, so we wanted to make sure we covered this important area in our review. In addition we have introduced a comment about the bias this word can produce in our review.

Line 355-356: Highest frequency of models?

Resolved in table 5.

Table 5: What is meant by machine learning (3%)?

Resolved in table 5.

Line 395: I would call I t Dairy research rather than dairy industry.

Done line 430.

Line 429-430: Are all the concepts identified focused on the reason for forecasting/predicting production? I find it difficult to see the figure (low resolution) but I am not convinced that everything that is put down relates to why it could be good to predict production?

Resolved on lines 465-470. The figures are also improved.

Line 528-539: The variability between published results will of course depend on input data quality, model validation methods, data set size and homogeneity and other factors. I am not sure what you want to say with this section. All the papers you have been using are published, perhaps that could also be discussed (is there a threshold on accuracy/R2/other measures to get published).

It is answered on lines 583-587.

558-563: Unclear writing (e.g. “in cases where what can be useful when…”)

It is answered on lines 603-607.

Reviewer 3 Report

Comments and Suggestions for Authors

An interesting article, could do with a little more justification of why it was done and the gap it fills 

Simple summary could be simplified further, artificial neural networks and convolutional neural networks are quite complex topics 

Abstract: IoT needs explaining or to appear in full first

Introduction:

L99 this does not make sense, check sentence and intended meaning 

Materials and Methods:

L153 - scoping rather than scope?

Is Web of Science and Scopus databases enough? Could you have looked at other sources? What about grey literature? 

What was justification for not English?

Results:

Figure 1 is useful but currently very blurry and not readable, can you improve the resoloution or is this an issue in being converted to pdf? 

L215 is this an extra full stop at the start?

Should this be described in the methods "bibliographic analysis, we have used specialized software like Bibliometrix and Vosviewer rather than the first mention being in the results 

L227 remove out and change point to points

L242 to 257 This whole section is basically discussion not results and should be moved 

Figure 2 again very poor quality 

Discussion:

Mainly fine could do with a little more on limitations of study 

Comments on the Quality of English Language

Does need some improvement and proof reading 

Author Response

An interesting article, could do with a little more justification of why it was done and the gap it fills 

It is answered on lines 101-106.

Simple summary could be simplified further, artificial neural networks and convolutional neural networks are quite complex topics 

The simple summary is corrected.

Abstract: IoT needs explaining or to appear in full firs

To solve this problem, the term IOT is removed from the abstract.

Introduction:

L99 this does not make sense, check sentence and intended meaning 

It is corrected on line 101.

Materials and Methods:

L153 - scoping rather than scope?

Corrected in line 164.

Is Web of Science and Scopus databases enough? Could you have looked at other sources? What about grey literature? 

It is answered online 651-656. It could be a limitation of the review, but we are confident we have detected the most relevant papers on this topic.

What was justification for not English?

It is answered on line 651-656.

Results:

Figure 1 is useful but currently very blurry and not readable, can you improve the resoloution or is this an issue in being converted to pdf? 

Improved resolution of the figures.

L215 is this an extra full stop at the start? Should this be described in the methods "bibliographic analysis, we have used specialized software like Bibliometrix and Vosviewer rather than the first mention being in the results 

Corrected in lines 222-226.

L227 remove out and change point to points

Line 248 is eliminated.

L242 to 257 This whole section is basically discussion not results and should be moved 

It is eliminated, it is present in the discussion.

Figure 2 again very poor quality

Figure is improved.

Discussion:

Mainly fine could do with a little more on limitations of study 

It is answered on lines 651-662.

Round 2

Reviewer 1 Report

Comments and Suggestions for Authors

---

Reviewer 3 Report

Comments and Suggestions for Authors

thank you for improving the manuscript